# OpenReview forum: "Enhancing Certified Robustness via Block Reflector Orthogonal Layers and Logit Annealing Loss"
_ICML.cc/2025/Conference — ICML 2025 spotlightposter_

### Official Review · Reviewer_8nWp · 2025-02-15

**Overall Recommendation:** 4

**Summary:**

Lipschitz neural networks are either trained or constructed such that their Lipschitz constant is small, enabling easy verification of the network to adversarial perturbations. Ways of obtaining networks with small Lipschitz constants include a) regularising the network at training time or b) designing layers which have a small Lipschitz constant by construction while maintaining expressiveness. This work introduces Block Reflector Orthogonal (BRO) layers which are a new type of orthogonal layer designed to have a Lipschitz constant of 1 using a new low-rank parameterisation. Such orthogonal layers are developed for both linear and convolutional operators. The authors further propose a new Logit Annealing (LA) loss for training Lipschitz neural networks which aims at balancing robustness considerations for different data points better by focusing on improving robustness for data points with a small margin. The evaluation on a number of datasets including the very large ImageNet shows that the new BRONet architecture outperforms existing approaches in terms of l2-certified and standard accuracy while being computationally efficient.

**Claims And Evidence:**

- The authors state that the BRO layers "unlock the potential of applying orthogonal layers to more advanced architectures". However, as far as I understand the paper they present an orthogonal formulation for dense and convolutional layers, both of which were already supported by other methods in the literature.
- The authors claim that they achieve state-of-the-art performance on a number of datasets which is mostly supported by the experimental results. However, especially for CIFAR10 and CIFAR100 in Table 1 the gains of the proposed method seems relatively small when compared to the literature. For TinyImageNet and ImageNet, the gains compared to the baselines are larger, but it should be noted that significantly fewer baselines were run for these datasets. In Table 4 the performance gain when using the proposed BRO backbone is relatively small in most cases.
- On large perturbations the performance of BRO seems to decline sharply (see Table 1), though this is acknowledged in the limitations section.

**Essential References Not Discussed:**

To the best of my knowledge, all relevant related works in the field of $\ell_2$ robustness certification are discussed by the authors. There is also a large body of work on networks that are robust to $\ell_\infty$ perturbations (such as [1-3]) which could briefly be discussed. Other works in the field such as [4] also discuss these and the relation of their work on $\ell_2$ robustness to $\ell_\infty$ robustness.

[1] Mao, Y., Müller, M.N., Fischer, M. & Vechev, M. (2023) TAPS: Connecting Certified and Adversarial Training. doi:10.48550/arXiv.2305.04574.

[2] Mueller, M.N., Eckert, F., Fischer, M. & Vechev, M. (2023) Certified Training: Small Boxes are All You Need. In: 1 February 2023 p. https://openreview.net/forum?id=7oFuxtJtUMH.

[3] De Palma, A., Bunel, R., Dvijotham, K., Kumar, M. P., Stanforth, R., & Lomuscio, A. (2023). Expressive Losses for Verified Robustness via Convex Combinations. arXiv preprint arXiv:2305.13991.

[4] Xu, X., Li, L. & Li, B. (2022) LOT: Layer-wise Orthogonal Training on Improving l2 Certified Robustness. Advances in Neural Information Processing Systems. 35, 18904–18915.

**Experimental Designs Or Analyses:**

The design of the experiments is sound, the approach is compared to all (to the best of my knowledge) relevant previous works on networks robust to perturbations in the $\ell_2$ norm and the evaluation is conducted across a variety of datasets which are normally used in the certified training literature.

**Methods And Evaluation Criteria:**

- Proposing layers with a small Lipschitz constant to obtain networks that are easier to certify is a well-known approach and therefore makes sense
- The benchmark datasets are standard ones that are used in a variety of previous works. The fact that the authors run experiments on Imagenet is commendable, given the substantial resources that are required for this.

**Other Comments Or Suggestions:**

Small typo: Line 68, right column: Assuming $f(x)$ is the output logits of a neural network --> Assuming $f(x)$ **are** the output logits of a neural network

Overall, the paper is well-written. One thing I noticed is that articles are often omitted in places where they should be present. Some examples:
- Line 90f: We construct various Lipschitz networks using BRO method --> We construct various Lipschitz networks using **the** BRO method
- Line 308: the annealing mechanism, draws inspiration from Focal Loss --> the annealing mechanism, draws inspiration from **the** Focal Loss
- Line 309: During training, LA loss initially promotes --> During training, **the** LA loss initially promotes
- Line 315: Consequently, LA loss allows Lipschitz models to learn --> Consequently, **the** LA loss allows Lipschitz models to learn
I did not write all of the cases in which this is done down, but to improve the reading flow it would be nice if the authors could do a round on the paper where they focus on eliminating these kinds of mistakes.

**Other Strengths And Weaknesses:**

A weakness I noticed is that there are a number of parameters which need to be tuned, for example, the LA loss introduces three parameters and for the BRO layer the rank $n$ needs to be chosen. The authors seem to explicitly tune this parameter on some datasets (see Table 9) while choosing certain values for other datasets. For example, it was unclear to me how the authors arrived at choosing $n=\frac{m}{2}$ for ImageNet but then at $n=\frac{m}{8}$ for TinyImageNet. The fact that the values of $n$ that are used differ not only between different datasets, but also between different network architectures (see Appendix D2) raises some concerns about how easy it would be to choose this parameter in practice without having to conduct a number of tuning runs.
The tuning of the three hyperparameters that the LA loss introduces seems somewhat unclear to me, an ablation study for $\beta$ is presented in Table 14 but for $T$ and $\xi$ the authors state that they "slightly adjust the values used in Prach & Lampert (2022) to find a better trade-off position" - how this is done remains unclear.

**Questions For Authors:**

1. Could the authors share any insights they might have regarding the performance of BRO on large perturbations? I find it surprising that other methods would suddenly perform so much better than BRO on these.
2. Do the authors have any heuristics or similar that they would use for selecting hyperparameters such as the rank $n$? How would they propose their training approach is used on new/unknown datasets or with modified variants of the BRONet architectures?
3. Could the way in which $T$ and $\xi$ were chosen be clarified by the authors?

**Relation To Broader Scientific Literature:**

The key contributions of the work, namely the BRO layer and the related network architectures, are related to a number of previous works on designing neural networks layers which exihibit small Lipschitz constants while also preserving sufficient expressivity. The BRO approach differs in that the layer itself is not a universal approximator, but the authors demonstrate that it is nevertheless expressive enough to outperform competing approaches in a number of cases. The orthogonal convolution layer introduced by the authors builds on a number of insights from the paper by Trockman and Kolter, but the specific parameterisation used for both the convolutional and the dense layers is novel.

**Theoretical Claims:**

I checked the proofs for the orthogonality of both the dense and convolutional layers proposed in the work (Appendix A1 and A2) which appear to be the most important contribution of the work. These look correct to me and I couldn't find any issues with them. The theoretical claims made in the paper appear sound to me. I found the Rademacher-complexity-based argument that is used to motivate the logit annealing loss function somewhat difficult to understand, but the theoretical analysis is accompanied by a more intuitive motivation which I find convincing.

---

> ### Author Rebuttal · Authors · 2025-03-30
>
> **Q: Regarding performance on large perturbations**
>
> BRONet indeed achieves the best performance on both clean and certified accuracy at $\varepsilon = 36/255$, but is less consistent for larger perturbations $\varepsilon$. Interestingly, we have observed that less expressive Lipschitz models tend to yield slightly higher certified accuracy at large $\varepsilon$, but at the cost of lower clean accuracy and certified accuracy at smaller $\varepsilon$. This tendency to fit only certain examples well to achieve large certified radii is not ideal, as our ultimate goal is to develop a robust model that maintains strong natural classification performance while simultaneously achieving favorable certified robustness as an additional benefit.
>
> ---
>
> **Q: Selecting the rank**
>
> For selecting the rank $n$, we recommend starting with $n = m/2$ and iteratively reducing it by half if needed, as supported by the experiments in Table 9. Intuitively, since $n$ determines the proportion of $+1$ and $-1$ eigenvalues in the orthogonal matrix $W$ (Proposition 1), an imbalanced distribution may reduce the diversity of $W$.
>
> ---
>
> **Q: Hyperparameters in the LA loss**
>
> We perform a hyperparameter grid serach around the values recommended by AOL (Prach & Lampert, 2022). Specifically, we evaluated the temperature $T \in \\{ 0.25, 0.5, 0.75, 1.0 \\}$ and the offset parameter $\xi \in \\{ 0.5, 1.0, 1.5, 2.0, 2.5, 3.0 \\}$ with LipConvNet-10-32 on CIFAR-100.
> When used on other datasets and architectures, these hyperparameters performed very well, so we did not fine-tune them further to reduce computational cost. We will include this information in Appendix D.4 in the revision.
>
> ---
>
> **Q: Regarding the statement "unlock the potential..."**
>
> We agree with your observation. Our intent was to highlight that BRO provides a promising parameterization that enhances robustness while reducing resource requirements for orthogonal layers. This, in turn, improves their applicability in more advanced architectures. We will revise the sentence for clarity.
>
> ---
>
> **Q: Performance in Table 1&4**
>
> For Table 4, we present an ablation study on different Lipschitz layers. The improvements are relatively small because all methods incorporate the proposed LA loss, which already narrows performance differences across backbones. Since Lipschitz networks have long faced scalability issues, there are still few baselines in the literature for TinyImageNet and ImageNet. Nevertheless, we would like to emphasize that combining both BRO and LA leads to a notable improvement on these more challenging datasets, outperforming the state-of-the-art LiResNet.
>
> ---
>
> **Q: Regarding typos**
>
> We appreciate your effort in identifying the typos. We have revised them accordingly.
>
> ---
>
> **Q: $\ell_\infty$-norm robustness**
>
> We appreciate your suggestion and will include a brief discussion on the literature regarding $\ell_\infty$-norm robustness.
>
> Additionally, we conduct empirical test against $\ell_{\infty}$ AutoAttack [1] on CIFAR-10, which we will provide it in the revision. The results for the $\ell_{\infty}$ certifed baselines are from the literature [2][3].
>
> | Method    | Clean     | Adv. Acc$\ell_{\infty}=2/255$ | Adv. Acc.$\ell_{\infty}=8/255$ | Certified Acc.                                           |
> | - | - | - | - | - |
> | STAPS $\ell_{\infty}=2/255$ | 79.75     | 65.91                            | N/A                              | 62.72 ($\ell_{\infty}=2/255$)                           |
> | SABR $\ell_{\infty}=2/255$  | 79.52     | 65.76                            | N/A                              | 62.57 ($\ell_{\infty}=2/255$)                           |
> | IBP $\ell_{\infty}=8/255$   | 48.94     | N/A                              | 35.43                            | 35.30 ($\ell_{\infty}=8/255$)                           |
> | TAPS $\ell_{\infty}=8/255$  | 49.07     | N/A                              | 34.75                            | 34.57 ($\ell_{\infty}=8/255$)                           |
> | SABR $\ell_{\infty}=8/255$  | 52.00     | N/A                              | **35.70**                        | 35.25 ($\ell_{\infty}=8/255$)                           |
> | BRONet-L (Ours)             | **81.57** | **68.76**                        | 21.02                            | 70.59, 57.15 , 42.53 ($\ell_{2}=36/255,72/255,108/255)$ |
>
> [1] Croce, et al. "Reliable evaluation of adversarial robustness with an ensemble of diverse parameter-free attacks." International Conference on Machine Learning (ICML), 2020.
>
> [2] Mueller, et al."Certified training: Small boxes are all you need." International Conference on Learning Representations (ICLR), 2023.
>
> [3] Mao, et al. "Connecting certified and adversarial training." Advances in Neural Information Processing Systems (NeurIPS), 2023.

---

> > ### Comment · Reviewer_8nWp · 2025-04-03
> >
> > Thank you for the rebuttal, I believe that all the questions that I raised were sufficiently addressed. While the performance gains achieved by BRO are probably not groundbreaking, I am aware of the fact that gains in this area are becoming increasingly difficult to achieve. I do think that this is a sound paper which proposes a novel approach that achieves better results, so in light of this, I will raise my score.

---

### Official Review · Reviewer_RtHN · 2025-03-10

**Overall Recommendation:** 4

**Summary:**

Lipschitz neural networks allow certified robustness without inference overhead; they are built by composing constrained layers. In this paper, the authors propose two improvements over the previous state of the art: they introduce a novel parametrization to construct orthogonal convolutions (BRO convolution), which aims to achieve a good performance/cost tradeoff thanks to a low rank parametrization that achieves orthogonalization without the need for an iterative algorithm. They also introduce a "Logit annealing loss function" that does not suffer the gradient issue of the previous losses.

**Claims And Evidence:**

This paper's approach performs well on standard benchmarks. It improved the previous state of the art by a decent margin. (However, an absolute comparison is subject to questions; see Supplementary material section.)

About LA loss:
Thanks to both theoretical and experimental work, the motivations behind LA loss are clear, and the experiments show a small but consistent improvement. This opens interesting perspectives about the properties of a good loss for robustness. Also, they explore the impact of the newly introduced hyper-parameter.

About BRO:
The extensive experiments cover many facets, such as speed/memory, robustness performance, and numerous ablation studies. This consequent work could be more impactful with a presentation centered on the paper's central claims. (See Experimental Designs Or Analyses)

**Essential References Not Discussed:**

I did not noticed any missing essential references.

**Experimental Designs Or Analyses:**

Experiments do not efficiently show that BRO reduces resource requirements. In particular, resource requirements might not correlate with the number of parameters, especially if  BRO is under-parametrized and based on FFT (which scales differently with respect to the number of channels and input size compared to usual convolution). The performance should be seen through a tradeoff between the number of parameters <-> performance <-> training cost.

Also, Cayley is missing in performance evaluation (Fig 2 and 5), which is surprising given the close proximity between the two methods (BRO can be seen as an under-parametrized version of Cayley).

**Methods And Evaluation Criteria:**

yes

**Other Comments Or Suggestions:**

typos:
- l761 Proposition -> lemma
- l753: define $F$ and $\bigotimes$

Suggestion:
- figure 2 uses the number of layers as x axis. This is pretty uninformative about how the layer scale ( as expected, it shows a linear increase for both runtime and memory, which is displayed in log scale for runtime). Especially when the displayed layers scale very differently with respect to the image size. I do expect BRO > SOC for small inputs like 16x16, but expect the opposite for large inputs like 224x224 (explaining why BROnet starts with a stem of an unconstrained convolution with stride 7 in imagenet)

**Other Strengths And Weaknesses:**

My main concern is the certificate correction discussed in the supplementary material. *I will raise my score if the authors show me the part of the code I may have missed or if their results have been corrected*. I think those results can be impactful even without the method being state-of-the-art.
My second main concern is about stating more clearly how the layer scales with respect to the number of channels, and input size, which would help a user select the adequate architecture.

Besides these two issues, I think this paper can be impactful for two reasons:
- it shows that under-parametrization can create a regime where the reduced resource cost can outweigh the reduced expressiveness.
- a careful analysis of the loss's gradient can lead to more effective losses for robustness

**Questions For Authors:**

I'm curious about the use of BRO parametrization for dense layers (which seems implemented in your code). Does the gain in terms of runtime outweigh the under-parametrization ?

**Relation To Broader Scientific Literature:**

The paper situates itself well within the current literature, clearly identifying its unique contributions relative to existing methods.

**Theoretical Claims:**

- ap A.4: although I agree with the reasoning, the padding must account for the fact that once a $k\times k$ kernel is orthogonalized in frequency domain, its spatial domain equivalent is not a $k\times k$ convolution anymore (it becomes an $s\times s$ convolution). Under this light, the input should be padded with $s$ value instead of $k$ values.
- Although the proof in A.1 is likely true, I did not understand the part showing that the parametrization results in real weights, especially since it is unclear how this conclusion is drawn from equation 16 ( line 792 ). An explanation of what $F$ and $\bigotimes$ are could alleviate this issue (equation 13 and line 754).

---

> ### Author Rebuttal · Authors · 2025-03-30
>
> **Q: Regarding Standardization and Certification**
>
> For the BRONet/LiResNet experiments, the dataloader outputs data in the [0,1] range without any additional standardization or normalization, as could be confirmed by the dataset functions in the `bronet/tools/dataset/` folder.
> For the LipConvNet experiments, the images are standardized in the dataloader, but the certification process has been adjusted to ensure that the reported budgets are correctly aligned with the [0,1] range. Specifically, as shown in line 202 of `lipconvnet/trainer/trainer.py`, `L = 1 / torch.max(std)`, which is then passed to `evaluate_certificates()` in line 265.
> We appreciate your diligence in reviewing the implementations and believe this explanation resolves your concerns.
>
> ---
>
> **Q: Regarding Scaling behavior and comparative analysis**
>
> We provide new runtime and memory comparison plots for LipConvNet-20 via the anonymous links below.
>
> [Link 1](https://ibb.co/9mJntHvw)
>
> [Link 2](https://ibb.co/HfsKGXbN)
>
> The x-axis represents the initial channels, while the y-axis shows runtime and memory consumption. The plots also include experiments with various input sizes, denoted by $s$. A missing dot indicates that an experiment encountered an out-of-memory error. The results demonstrate the scalability of BRO with respect to both channel and input size. We will include the figures in the next revision to help users select an adequate architecture. The experimental settings are consistent with those used in Figure 5 of the main paper.
>
> We also include the results of Cayley layer in the revised Figure 2 via the below anonymous link.
>
> [Link 3](https://ibb.co/jYD2Gm1)
>
> ---
>
> **Q: Regarding zero-padding**
>
> Thank you for pointing this out. We will revise our description on line 214. For clarity, the current implementation reduces the effect of convolution across edges but does not fully eliminate it. Since zero-padding does not affect the validity of the certification, one may also consider applying it post-orthogonalization, albeit at an increased computational cost. We will include this note in the revision..
>
> ---
>
> **Q: Regarding the proof in A.1**
>
> We apologize for the unclear notations and have revised them. For $\mathcal{F}\_c = S_{c, s^2} \left(I\_c \otimes (F \otimes F)\right)$ (Line 754), $F$ is the DFT matrix, and $\otimes$ is Kronecker product.
> Regarding Equation (13), it is a result from Trockman & Kolter (2020), showing that any 2D circular convolution operation can be block-diagonalized by a Fourier-related matrix $\mathcal{F}.$
>
> As for Equation (16), it is intended to explicitly demonstrate the orthogonality of BRO and its real-preserving property. The right-most term explicitly shows that the BRO convolution we use is orthogonal, as it consists of three unitary matrices. The middle term shows that $\text{BRO}(C)$ is real if $C$ is real-valued, because that $\text{BRO}(\cdot)$ applied to a real matrix always produces a real matrix. As we parameterize $C$ in the real domain, the output of the BRO convolution is guaranteed to be real-valued.
>
> ---
>
> **Q: Regarding parameterization for dense layers**
>
> For ConvNets, the computational cost is primarily dominated by convolution operations, so we did not find much advantage in swapping out the dense layers. That being said, it could be worth exploring in other architectures that rely more heavily on large dense layers and could benefit from the properties of orthogonal parameterizations.

---

> > ### Comment · Reviewer_RtHN · 2025-04-02
> >
> > The author's answer alleviated my main concerns (mainly about the certificate correction). I will raise my score.

---

### Official Review · Reviewer_XtW5 · 2025-03-21

**Overall Recommendation:** 4

**Summary:**

This paper introduces a new 1-Lipshitz layer using the Block Reflector Orthogonal (BRO) parameterization of low-rank orthogonal matrices for constructing image classifiers with certified robustness. In addition a new logit annealing loss function is developed to balance margin learning across data points, addressing the limited capacity issue inherent in Lipschitz networks. The resulting architecture, BRONet, is demonstrated to achieve state‐of‐the‐art l2 certified robustness on benchmarks including CIFAR‑10, CIFAR‑100, Tiny‑ImageNet, and ImageNet.

**Claims And Evidence:**

The paper claims significant gains in certified robustness and computational efficiency over prior methods (e.g., LOT, SOC) through the novel BRO layer and LA loss.

To my knowledge, the use of the block reflector parameterization has not  been used before for 1-Lipschitz layers.

Evidence includes runtime and memory comparisons, as well as ablation studies showing marginal increases (around 1–2 percentage points) in certified accuracy.

Although the improvements are modest, the architecture is technically sound and empirical evidence is quite thorough.

**Essential References Not Discussed:**

I think they have covered the literature of 1-Lipschitz layers and certified robustness well.

**Experimental Designs Or Analyses:**

The experimental design is comprehensive, with comparisons across multiple architectures and datasets, including ablation studies that assess the effect of the LA loss.

Regarding runtime shown in Figure 2, I would also be curious to compare the runtime and memory efficiency of methods which do not require matrix inversion in their forward pass such as AOL or SLL. They may not be as competitive for certified accuracy, but will scale better.

**Methods And Evaluation Criteria:**

The main application of 1-Lipschitz layers is certified robustness of classifiers using a margin argument.

The method is evaluated on benchmarks for certified robustness of image classification problem with respect to the l2-norm perturbation threat model. Although this l2 threat model is not very practical as model for modeling adversarial attacks in images, it is widely studied in the certified robustness literature. Other common models such as l1 and l-infinity threat models are not addressed by this work.

**Other Comments Or Suggestions:**

None

**Other Strengths And Weaknesses:**

Strengths:

The I think the low rank orthogonal architecture is a nice way to balance efficiency and performance, avoiding iterative schemes of SOC and LOT to perform matrix inversion.

The theoretical justification behind the LA loss is fairly interesting and seems to make improvements experimentally in most cases. I can see this loss being a common ablation in future works on certified robustness.


Weaknesses:

BRO still seems to have a significant computational overhead when computing the sparse matrix inverse that may limit its scalability.

Although making even incremental improvement in certified l2 robustness (as many other papers have pursued) is challenging, there is not really anything paradigm shifting here. These parameterizations still perform very poorly on CIFAR100 and ImageNET clean accuracy, where 50% accuracy is far from acceptable in practice. I hope the community will begin to consider more realistic threat models and benchmarks as even the maximum l2 perturbation size considered $\epsilon = 108/255$ can not saturate a single pixel.

**Questions For Authors:**

How sensitive is the LA loss to hyperparameter settings across different architectures? It seems $\beta=5$ is used in all benchmarks. I see in Table 14 that this is about optimal for CIFAR100, but it is generally best across the other datasets as well? How do you recommend tuning these parameters?

**Relation To Broader Scientific Literature:**

The work builds on established concepts such as orthogonal parameterizations (e.g., via the Cayley transform) and Lipschitz network designs. The paper does a pretty good job of contextualizing and benchmarking many other relevant methods in literature (figure 1 is quite nice in this regard). Introducing the low-rank block reflector approach and a new loss function, which are valuable incremental contributions. Additionally, I can see the LA loss becoming a standard ablation in the area of certified robustness (I say ablation because sometimes the CR loss is still better).

However, many other papers have pursued similar incremental improvements of certified robustness in these l2-norm threat models and the overall contribution is not paradigm shifting. Its not easy to make even small improvements in certified robustness, but I hope the community will begin to consider more realistic threat models as image classifiers with a clean accuracy of 50% are not very useful.

**Theoretical Claims:**

The main theoretical claims of the architecture are based on well-established parameterizations of orthogonal matrices which I believe to be correct.

The risk bound derived in Theorem 1 and Proposition 3 seem to be correct and based on established results and are used to motivate the design of the logit-annealing (LA) loss. In particular it justifies the intuition that we should not maximize the margin uniformly in the presence of Lipschitz constraints. I think this is a good motivation for the LA loss.

I do think the proof for Theorem 1 could be stated more clearly in a proof environment or in a subsection "proof of Theorem 1", I couldnt find a proof for Proposition 3 in the paper. Even if it is immediate from previous work (I assume its Ledoux and Talagrand 2013), its worth stating briefly. If its being cited from another paper, it would be better to cite the result explicitly.

---

> ### Author Rebuttal · Authors · 2025-03-30
>
> **Q: Regarding Theorem 1 and Proposition 3**
>
> Thank you for your feedback. We will enhance the clarity of Theorem 1 by presenting its proof in a dedicated subsection. For Proposition 3, it is from (Ledoux and Talagrand 2013) and we will explicitly cite it in the statement for clarity.
>
> ---
>
> **Q: Regarding Figure 2**
>
> We appreciate the reviewer's curiosity. Indeed, 1-Lipschitz non-orthogonal layers could offer better computational efficiency but less competitive for certified robustness. As shown in Table 1, the large-scale SLL models do not achieve the same level of certified accuracy. This highlights the limitations of these layers when it comes to scaling for robustness.
>
> ---
>
> **Q: The hyperparameters in the LA loss**
>
> We selected the hyperparameters in the LA loss by testing them on LipConvNet-10-32 and applied the same values across all experiments. When used on other datasets and architectures, we found these hyperparameters to perform well. Therefore, we did not fine-tune them further in order to reduce computational cost.  In practical applications, users can further refine the hyperparameters using grid search or random search, which may lead to even better results.

---

> > ### Comment · Reviewer_XtW5 · 2025-04-07
> >
> > My concerns have been addressed. I have adjusted my score accordingly.

---

### Official Review · Reviewer_oDFR · 2025-03-22

**Overall Recommendation:** 3

**Summary:**

The paper proposes a new method to construct 1-Lipschitz neural networks, namely, the L2 norm Lipschitz constant for each layer is 1. A 1-Lipschitz network is very useful for guaranteeing the robustness of neural networks. The paper claims to outperform existing 1-Lipschitz network designs such as SOC and Cayley layers.

**Claims And Evidence:**

The author claims that the proposed parametrization for 1-Lipschitz neural networks is efficient and outperforms existing methods. Experiments are conducted on multiple datasets, multiple perturbation epsilons, and multiple baseline methods.

**Essential References Not Discussed:**

Essential references are discussed and experimentally compared.

**Experimental Designs Or Analyses:**

The experimental design follows existing work in this field: clean accuracy and certified accuracy comparisons on a few datasets on different perturbation epsilons. Many baselines are included. The work in general performs well, although the improvements are relatively marginal and inconsistent in some settings.

**Methods And Evaluation Criteria:**

The main method is based on the block reflector construction of orthogonal matrices. This construction is not discussed in prior works (under the context of 1-Lipschitz networks), although some techniques used are similar (e.g., the use of FFTs for orthogonal convolution). The authors also proposed to adjust the commonly used cross-entropy loss for 1-Lipschitz networks, inspired by its limited capacity to learn large margins.

**Other Comments Or Suggestions:**

None

**Other Strengths And Weaknesses:**

The main strength of the paper is the novel formulation of 1-Lipschitz neural network layer, and the adjusted cross-entropy loss for 1-Lipschitz training.

My biggest confusion is about the evaluation - reading the paper, it sounds like Table 1 includes BRO layers only and does not mention LA loss. However, appendix B.5 mentioned that LA loss is used in all BROnets models. The LA loss is a generic approach that can be applied to any 1-Lipschitz network approach - it does not require any special features of BRO. Technically, it can be added to any existing Lipschitz approach, and it is uncertain that in Table 1 the major improvements come from the BRO, or actually the LA loss. Especially there are many newly introduced hyperparameters, such as the rank and the logic annealing hyperparameters. It is possible that the LA loss is the main contributor and the BRO may not outperform existing parameterization in a fair setting, since it is not a universal approximation.

To support the claim that the “expressive power of deep neural networks constructed using BRO is competitive with that of LOT and SOC” (line 243), we hope to see Table 1 listing both BRO and BRO + LA loss numbers.

**Questions For Authors:**

Unlike many prior works on orthogonal layers, the construction presented in this paper is low-rank. Is a low rank necessary for this parameterization? For the scenarios where we need more model capacity, how can we get a full-rank orthogonal matrix?

**Relation To Broader Scientific Literature:**

The design and creation of 1-Lipschitz network can be an important building block for creating neural networks with provable guarantees such as robustness and safety. In general, the techniques proposed by this may be valuable for the scientific community.

**Theoretical Claims:**

Rademacher complexity is used to prove that the capacity of 1-Lipscitz network is limited and may not fit large margins well. The analysis is based on the standard learning theory approach, and inspires the development of the LA loss.

---

> ### Author Rebuttal · Authors · 2025-03-30
>
> **Q: Regarding Table 1 Description**
>
> Thank you for your feedback. Table 1 indeed presents the combined results with the LA loss, as stated in the Appendix. To ensure clarity, we will explicitly indicate this by adding a (+LA) notation in the revised version.
>
> ---
>
> **Q: Fair Comparison of Different Parameterizations**
>
> We understand your concerns about fairness in comparison. To clarify, our comparisons in Table 2 (LipConvNet Benchmark) and Table 4 (Backbone Comparison) are conducted under fair conditions, with all baselines incorporating the LA loss. To ensure transparency and avoid any confusion, we will explicitly state this in the table descriptions.
>
> ---
>
> **Q: Regarding Low-rank parameterization**
>
> The orthogonal matrix $W$ in BRO ($W=I-2V(V^TV)^{-1}V^T$) is consistently full-rank, and the trainable parameters $V$ are constrained to be low-rank to prevent $W$ from degenerating into a negative identity matrix, as shown in Proposition 1. To increase model capacity, experiments (Appendix D.1, Table 9) suggest that enlarging $W$ and setting $V$ to half its rank improve data fitting.

---

> > ### Comment · Reviewer_oDFR · 2025-04-03
> >
> > Thank you for answering my questions. My main concern is still about the fair comparison, where we ideally want to present the main table (Table 1) with LA loss on and off. That will truly compare the effectiveness of the proposed parameterization and give readers a true understanding of the expressiveness of the proposed parameterization.
> >
> > The results in Table 2 and Table 4 are too limited and may not sufficiently support the claim that BRO is better than existing methods, since the benefits may come from LA loss, which can be directly combined to many existing methods.
> >
> > Thus, I believe this submission is borderline and cannot strongly support its acceptance.

---

> > > ### Author Response · Authors · 2025-04-03
> > >
> > > **Table 1 Revision**
> > >
> > > We appreciate your concern and have taken steps to provide a more comprehensive clarification. To explicitly address this, we will include the results both with and without the LA loss in Table 1. The performance of BRONet, both with and without LA, is presented below alongside the baseline LiResNet for comparison. On average, BRONet and LA contribute improvements of +0.32/+0.15 (CIFAR-10), +0.22/+0.57 (CIFAR-100), +0.72/+1.67 (Tiny-ImageNet), and +0.97/+1.47 (ImageNet), respectively, across different evaluation metrics.
> > >
> > >
> > > | CIFAR-10           | Clean    | $\ell_{2}=36/255$ | $\ell_{2}=72/255$ | $\ell_{2}=108/255$ |
> > > | ------------------ | -------- | ----------------- | ----------------- | ------------------ |
> > > | LiResNet           | 81.0     | 69.8              | 56.3              | 42.9               |
> > > | **BRONet-L**       | 81.0     | 70.2              | 57.1              | **43.0**           |
> > > | **BRONet-L (+LA)** | **81.6** | **70.6**          | **57.2**          | 42.5               |
> > >
> > > | CIFAR-100          | Clean    | $\ell_{2}=36/255$ | $\ell_{2}=72/255$ | $\ell_{2}=108/255$ |
> > > | ------------------ | -------- | ----------------- | ----------------- | ------------------ |
> > > | LiResNet           | 53.0     | 40.2              | 28.3              | 19.2               |
> > > | **BRONet-L**       | 53.6     | 40.2              | 28.6              | 19.2               |
> > > | **BRONet-L (+LA)** | **54.3** | 40.2              | **29.1**          | **20.3**           |
> > >
> > > | Tiny-ImageNet    | Clean    | $\ell_{2}=36/255$ | $\ell_{2}=72/255$ | $\ell_{2}=108/255$ |
> > > | ---------------- | -------- | ----------------- | ----------------- | ------------------ |
> > > | LiResNet         | 40.9     | 26.2              | 15.7              | 8.9                |
> > > | **BRONet**       | 40.5     | 26.9              | 17.1              | 10.1               |
> > > | **BRONet (+LA)** | **41.2** | **29.0**          | **19.0**          | **12.1**           |
> > >
> > > | ImageNet         | Clean    | $\ell_{2}=36/255$ | $\ell_{2}=72/255$ | $\ell_{2}=108/255$ |
> > > | ---------------- | -------- | ----------------- | ----------------- | ------------------ |
> > > | LiResNet         | 47.3     | 35.3              | 25.1              | 16.9               |
> > > | **BRONet**       | 48.8     | 36.4              | 25.8              | 17.5               |
> > > | **BRONet (+LA)** | **49.3** | **37.6**          | **27.9**          | **19.6**           |
> > >
> > >
> > > We believe that these, combined with Table 2 and 4 (comparing with baselines all using the LA loss), should help the reader fully understand the expressiveness of the proposed parameterization, independent of the LA loss.

---

### Decision · Program_Chairs · 2025-05-01

**Decision:**

Accept (spotlight poster)

**Comment:**

This paper introduces BRO layer that parameterizes a group of orthogonal (and thus Lipschitz-bounded) linear and convolutional layers. The paper also proposes logit annealing loss to encourage sufficiently large margins for more data points. When combined with diffusion-based data augmentation, BRONet achieves state-of-the-art certified L2 robustness on CIFAR10 and CIFAR100.

Most concerns were resolved after the discussion phase. All reviewers favor acceptance due to the solid contributions, but don't nominate for oral presentation due to some remaining concerns about the the small margin compared to baselines, the nontrivial computational overhead, and the limitations of the l2-certified robustness threat model.

Please make sure to address reviewers' comments in the camera-ready version. Specifically, please revise Table 1 according to the reviewer's comment to clarify the comparison with and without LA loss, fix the notation ambiguity in proof in A.1, and add runtime and memory comparison with Cayley.